# Activity and Pool Use in Relation to Temperature and Water Changes in Zoo Hippopotamuses (*Hippopotamus amphibious*)

**DOI:** 10.3390/ani10061022

**Published:** 2020-06-12

**Authors:** Eduardo J. Fernandez, Martin Ramirez, Nancy C. Hawkes

**Affiliations:** 1School of Behavior Analysis, The Florida Institute of Technology, Melbourne, FL 32901, USA; 2Woodland Park Zoo, Seattle, WA 98103, USA; martin.ramirez@zoo.org (M.R.); nancy.hawkes@zoo.org (N.C.H.)

**Keywords:** exhibit use, hippo, hippo enclosures, pool use, pool temperature, water change, welfare, zoo

## Abstract

**Simple Summary:**

The following study examined the behaviors and water vs. land use of an outdoor exhibit by three zoo hippos. Behavioral observations were correlated with water temperatures at the exhibit, and compared to the number of days (0, 1, or 2 days) since the water within the outdoor pool had been dumped and refilled. The water changing of the pools had little effect on either the behaviors or pool use itself by the hippos. Water temperatures affected both the behaviors and pool use by the hippos, with warmer water temperatures being directly correlated with greater activity and overall pool use. The results suggest that temperature, rather than water change, was the more important factor in increasing overall pool use and general activity for the exhibited hippos.

**Abstract:**

In the wild, hippopotamuses spend much of their daily activity in the water. In zoos, it is less clear the extent to which hippos spend time in the water. We examined how much time Woodland Park Zoo’s three hippos spent in their outdoor pool, based on: (a) temperature of the pool water, and (b) when the pool water was changed (approximately three times a week). Several digital temperature data loggers collected water and air temperature readings once every hour for six months. We correlated the water temperature readings with several behaviors the hippos could engage in, where the hippos were on exhibit (pool vs. land), and how many days it had been since a dump (0, 1, or 2 days). The results indicated that water changes had little effect on pool usage, while increasing water temperatures resulted in both increased activity and pool use. The results are discussed in terms of how these findings relate to wild hippo activity, current knowledge of zoo-housed hippo welfare, and future directions for zoo-housed hippo welfare and research.

## 1. Introduction

The common hippopotamus (*Hippopotamus amphibious*) is a large, semi-aquatic ungulate native to sub-Saharan Africa [1]. Hippos spend much of their day in the water, forming large groups, or pods, averaging 10–13 hippos per pod, but numbering up to 100 individuals [2,3,4]. While hippos spend much of their day in the water, most of their night activity consists of foraging for grass and other vegetation on land [1,5]. This nocturnal foraging can result in increased human–hippo conflicts (e.g., crop destruction and physical threats), and is therefore detrimental for both hippos and humans [6]. Hippos have also been viewed as an invasive species, having been artificially introduced to rivers in Colombia, which creates greater potential for human–hippo conflicts [7,8].

In zoos, the regular activity of exhibited hippos is less clear. While past publications have focused on the exhibit design and care of zoo hippos and other ungulates [9,10,11], only a handful of studies have examined the behaviors of zoo-housed hippos. Blowers et al. [12,13] examined both the social structure and exhibit use of zoo hippos, demonstrating preferences for both familiar individuals and water areas within an exhibit. Fazal et al. [14] examined the behavioral effects of introducing a female hippo to a solitary male hippo, showing increased activity as a result of the introduction. Little else has been done to quantify what hippos do in their zoo exhibits. A recent study suggests that North American zoos may not be adequately meeting the behavioral welfare of their hippos [15]. For instance, Tennant et al. [15] found that, contrary to wild hippo pod sizes of ≥ 10 and their regular nocturnal foraging activity, only a third of the zoos surveyed housed their hippos in groups of three or more, and almost half of the facilities surveyed limited nocturnal foraging opportunities for their hippos. Thus, the need for more empirical information on hippo activity in zoos is clear.

The following study examined the effects of water changes and temperatures on the activity of three zoo-housed hippos. Prior to the implementation of the study, it was hypothesized by care staff that both water changes of an outdoor pool and seasonal temperature variation affected the pool use and general activity of the exhibited hippos. We, therefore, examined the effects of: (1) days since the water was changed in the outdoor pool, and (2) air and water temperatures at the exhibit on (a) several possible behaviors observed, and (b) pool vs. land use within the exhibit. The purpose of the study was to document these behavioral effects, which would allow us to empirically evaluate the behavioral welfare of the exhibited hippos, as well as suggest future directions for the assessment and care of these and other zoo-housed hippos.

## 2. Materials and Methods

### 2.1. Subjects and Setting

Three zoo-born hippopotamuses (*Hippopotamus amphibious*) were the subjects of the study. Gertie, a 47-year-old, ~1500 kg female hippo at the start of the study, came from an unknown location and resided at the Woodland Park Zoo (Seattle, WA, USA) since January of 1966. Water Lily (“Lily”), a 31-year-old, ~1300 kg female hippo at the start of the study, was born at the Houston Zoo and resided at the Woodland Park Zoo since October of 1979. Guadalupe (“Lupe”), a 10-year-old, ~1530 kg female hippo at the start of the study, was born at Disney’s Animal Kingdom and resided at the Woodland Park Zoo since July of 2003.

All three hippos resided in an exhibit that contained three areas: An outdoor land area, ~1375 m^2^, an outdoor pool, ~650 m^2^, that held ~180 kL of water, and an indoor barn, ~300 m^2^ (see Figure 1).

The outdoor land area consisted of natural trees, grass and rocks, and browse was frequently provided in this area, as well as daily feedings. During the winter months (December–March), a heated pad built into the land area concrete, measuring ~4 m × 4 m, was turned on. The pad was heated to 15–20 °C. The outdoor pool was non-filtered, non-heated, and the water was changed (dumped and refilled) 3 times per week (see Procedures). Devices such as boomer balls and feeding enrichment were regularly delivered in the outdoor pool area. The indoor barn consisted of three individual stalls (4.5 m × 3 m, 4.5 m × 4 m and 4.5 m × 7 m), as well as a misting system. Browse, enrichment, and one of the daily feedings, were regularly provided in the barn.

The hippos were typically moved from the barn area to the outdoor exhibit by 09:30 h, and moved back into the barn at 18:00 h, with some variability depending on the weather and time of year. Diets for the hippos varied, based on both the individual and time of year, with 1300–1650 kcals consumed per hippo per day. All three hippos’ diets consisted of Mazuri^®^ (St. Louis, MO 63166, USA) ADF-25 herbivore diet, romaine lettuce, celery, cucumbers, melons, carrots, apples, rotational vegetables (root vegetables, onion, squash, and green peppers) and ad lib grass hay. Diets were provided to the animals several times a day, with a morning outdoor land area feeding at ~09:30 h, another outdoor land area feeding at ~15:00 h, and the rest of their diet being provided in the barn at ~18:00 h. 

### 2.2. Materials

Materials included three ThermoWorks^TM^ (American Fork, UT 84003, USA; thermoworks.com) USB-1 digital temperature data loggers (#TW-USB-1) and three protective metal cases (#TW-USB-CASE) for the temperature data loggers. The data loggers had an IP67 waterproof rating with a range from −35 to 80 °C (±1 °C). Other materials included Palm^®^ (Sunnyvale, CA 94085, USA) handhelds used to record behavioral data, an Event-PC program that was run on the Palm^®^ handhelds and designed specifically for this experiment by the Dr. James C. Ha at the University of Washington, and a notebook used to record potential errors and additional observations/field notes that occurred during a session.

### 2.3. Data Collection and Procedure

Prior to its implementation, the study was approved through Woodland Park Zoo’s Research Committee as well as the University of Washington’s Institutional Animal Care and Use Committee (IACUC #2858-06). Following approval, the three temperature data loggers were placed at three different points of the exhibit that did not receive direct sunlight: one was fixed to a tree above the exhibit and out of reach from both visitors and the hippos, and that was shaded by leaves and branches (air temperatures); another inside a floating boomer ball in the pool (surface water temperatures); and the third fixed inside the drainage grating at the bottom of the pool (deep water temperatures). The data loggers took hour-to-hour temperature data and were only removed upon completion of the study. Both surface and deep water temperatures were combined and averaged for a single hourly measure of water temperature. 

An ethogram consisting of eight behaviors was also developed prior to the implementation of the study (see Table 1):

The behaviors observed were mutually exclusive, and the inclusion of the “Other” observation category made the ethogram exhaustive. All 8 responses could be recorded in the pool, and 7 of the 8 responses could be recorded on land, with the exception of the “Submerged” response, which identified a hippo in the pool without the ability to identify the exact response. A modified scan sampling procedure [16] was used to record behaviors and exhibit location (land or pool) during all observation sessions. The number of hippos on exhibit (2 or 3 hippos) was recorded for behavior and exhibit location every 30 s for 0.5 h of observation for each session. These observations were then averaged for each session based on the total number of hippos engaging in each behavior in each area, and correlated with the air and water temperatures recorded during that session, as well as how recently the pool water was changed (0, 1, or 2 days).

All observations were conducted between 09:30–18:00 h, seven days a week, between 12 January, 2010 and 10 July, 2010 (493 total sessions across the 179 days [1–8 observations per day] for 246.5 h of behavioral observation). Temperatures were recorded from 6 January, 2010 to 11 July, 2010, each hour for a full 24-h day (187 days: 4488 hourly recordings). Observers were typically registered for independent research credit through the Psychology Department at the University of Washington (PSY 499) and received observation training by live training sessions at the beginning of each semester and weekly lab meetings throughout the study. Observations were examined weekly by the first author for consistency across all observers, and drift was accounted for during these weekly checks, as well as through weekly lab meetings. Observers were blind to the conditions of the study, although they had a general sense of the weather conditions during their observation sessions. A total of 33 observers collected behavioral data for the entire study. 

### 2.4. Statistical Analyses

SigmaStat™ (San Jose, CA 95131, USA) 11.0 was used to run all the statistical analyses. All tests failed the Shapiro–Blowers–Wilk tests for normality. Therefore, Spearman’s rank correlation (*r_s_*) was used to test the relationship between daily air and water temperatures. In addition, the percentage of water and land use across water change conditions (0, 1, or 2 days) and water temperatures (split into 4 quartile conditions), as well as differences in behavioral observations across water temperature conditions, were tested using a Kruskal–Wallis analysis of variance (ANOVA) on ranks. When significant differences (*p* < 0.05) for the ANOVAs were found, post-hoc pairwise comparisons (using Dunn’s Method) were implemented.

## 3. Results

Figure 2 shows the daily temperatures from 6 January, 2010 to 11 July, 2010, as well as the hour-to-hour temperature for the months of January and July. There was a strong positive correlation between daily air and water temperatures, *r_s_* (185) = 0.846, *p* < 0.001. The hour-to-hour temperatures showed more fluctuations in July compared to January, and between air compared to water temperatures, with the air and water temperatures in July ranging from 13.58–21.14 °C and 17.41–18.89 °C, respectively, and the air and water temperatures in January ranging from 6.18–9.30 °C and 7.73–8.15 °C, respectively.

In terms of overall activity throughout the study, 5 of the 8 behaviors occurred more than 1% of the time: Lying Down, 43.09% (*SE* = 1.94); Standing, 23.95% (*SE* = 1.32); Foraging, 16.47% (*SE* = 1.44); Locomotion, 7.85% (*SE* = 0.51); and Submerged, 7.12% (*SE* = 0.60). Therefore, statistical tests and graphs were only completed for these five behaviors. Overall water (pool) use occurred for 37.73% of the total time (*SE* = 0.89), and land use occurred for 62.27% of the total time (*SE* = 1.27).

Figure 3 shows the water vs. land use of the hippos based on days since a water change (0, 1, or 2 days). No significant differences were observed between the three conditions (Water Use: *x*^2^_2_ = 0.565, *p* = 0.754; Land Use: *x*^2^_2_ = 0.566, *p* = 0.754), with 2 days since a water change showing the greatest pool use (*M* = 41.47%, *SE* = 6.06). Similarly, no significant changes in the behaviors were observed in relation to the days since a water change: Lying Down, *x*^2^_2_ = 4.886, *p* = 0.087; Standing, *x*^2^_2_ = 2.543, *p* = 0.280; Foraging, *x*^2^_2_ = 5.269, *p* = 0.072; Locomotion, *x*^2^_2_ = 0.340, *p* = 0.844; and Submerged, *x*^2^_2_ = 0.856, *p* = 0.652.

Figure 4 shows the five most frequently occurring behaviors as well as water vs. land use across four different water temperature conditions: <10, 10–12, 13–15, and ≥16 °C. For the behaviors, there were only two statistically significant differences observed: Locomotion, *x*^2^_3_ = 9.759, *p* = 0.021, and Submerged, *x*^2^_3_ = 14.568, *p* = 0.002. For both Locomotion and Submerged, post-hoc tests showed a significant increase from 13–15 °C to ≥ 16 °C. (*p* < 0.05 for both). The three other behavior tests failed to find a statistically significant difference: Lying Down, *x*^2^_3_ = 6.539, *p* = 0.088; Standing, *x*^2^_3_ = 2.922, *p* = 0.404; and Foraging, *x*^2^_3_ = 2.590, *p* = 0.459. For water and land use, there were statistically significant differences for both tests (*x*^2^_3_ = 14.915, *p* = 0.002 for each). Post-hoc tests showed a significant increase in pool use and decrease in land use between the <10 °C and ≥16 °C conditions (*p* < 0.05 for both).

## 4. Discussion

### 4.1. Measuring Temperature

As expected, there was a strong correlation between air and water temperatures recorded at the exhibit. Nonetheless, this was important to document since accurate recordings of both air and water temperatures at the exhibit and during each observational session were critical to the success of the study. Also, demonstrating that water temperatures were more stable than air temperatures across both hours and days allowed us to focus on the link between the behaviors observed and the recorded water temperatures.

### 4.2. General Activity and Pool Use

The hippos spent almost half their time resting (i.e., Lying Down), and almost the other half their time in general activity (i.e., Locomotion, Foraging, or Standing). This activity pattern is similar to what has been reported for hippos in the wild, with close to half their time resting, a third of their time moving, and a fifth of their time foraging [17]. Likewise, the hippos in this study spent over a third of their time in the water, with their wild counterparts reportedly spending approximately half their time in the water [1]. Factors such as temperature seem to directly influence the total time spent in the water for the hippos in this study, which has been similarly reported for wild hippos ([18]; see below). Interestingly, the hippos in this study also showed an unreported spike in pool use just before the 15:00 h land feeding, followed by increased land use during the feed. It is not clear why this increase in pool use would occur, either as a behavioral contrast effect due to the following increased land use while foraging, or possibly as anticipatory feeding behavior related to their natural foraging repertoire [19,20]. Regardless, the behavior of these three exhibited hippos appeared similar to their wild counterparts, both in terms of general activity and overall time spent in the water.

### 4.3. Water Change Effects

No significant differences were observed in either pool/land use or general activity as a result of the amount of time (days) since the pool water was changed. This result differed from anecdotal reports by the care staff, who suggested that the hippos used their pools less as more time passed since a water change. This was also one of the primary reasons for examining the effects of water changes on both the hippos’ general activity and their overall pool use.

It is worth noting that the quality of the water was not directly examined, but rather, amount of time (days) since the last water change, and that anecdotal implications by the care staff were that the hippos used their pool less when the water quality was diminished. Nonetheless, decreased water quality, based on the amount of time during which 2–3 hippos spent in an unfiltered pool, is a relatively safe assumption, and thus, the reduced quality of the water up to two days since a water change appeared to have little effect on the activity or pool use of the hippos.

### 4.4. Temperature Effects

Most of the significant effects observed were in relation to the temperature of the water, with locomotion and pool use showing their highest percentage of occurrence and land use, showing its lowest percentage of occurrence in the ≥16 °C water condition. While not statistically significant, it is additionally worth noting that the highest percentage of occurrence for foraging and standing, as well as the least percentage of occurrence for lying down, were also observed in the ≥16 °C water condition.

While these results suggest a direct relationship between the pool water temperatures and the activity and pool vs. land use of exhibited hippos, it is important to note that this is a correlational study, not a causal one. For instance, it is not clear whether pool use increased because water temperature increased, or whether water use increased because air temperature increased (i.e., escape from heat). Such causal relations are beyond the scope of this study, as identification of such functional variables would likely require experimental manipulation of water and/or air temperatures. However, examinations of hippos in the wild have demonstrated that hippos modify their time spent in the water as a form of thermoregulation, increasing or decreasing their time spent in the water based on both overall sun exposure and increasing land temperatures, as well as to escape colder water temperatures [18]. Outdoor zoo exhibits for hippos should be sensitive to both possibilities and consider the needs of hippos to use their pools to thermoregulate during both hot and cold periods.

Future research should consider the behavioral welfare implications of pool use, as well as the possibility of increasing both pool use and general activity through events such as pool temperature regulation, environmental enrichment, and husbandry training. Preferences for pool and land temperatures could be directly assessed through preference assessments, which could give hippos choices in exhibit spaces used based on such area temperature differences [21,22,23]. Likewise, other studies have investigated the use of training interactions combined with enrichment to increase time spent swimming in zoo-housed penguins [24]. Future research could examine similar effects with zoo hippos, particularly how factors such as enrichment relate to the amount of time hippos spend in enclosure water areas and/or pools. To promote the best welfare for zoo-housed hippos, zoos should attend closely to hippo pool use activity. We hope this study is just one step in attending to such data-driven detail.

## 5. Conclusions

The results of this study suggest that attending to water and/or general exhibit temperatures may be one of the most critical factors when exhibiting hippos. In our study, amount of time since a water change, and therefore water quality, appeared to have little effect on how much the hippos used their pools. However, the amount of time the hippos spent being active and using their pools was positively correlated with the water temperature. Management practices for exhibited hippos should consider hippo pool use in relation to hippo pool temperature and modify exhibit spaces accordingly.

## Figures and Tables

**Figure 1 animals-10-01022-f001:**
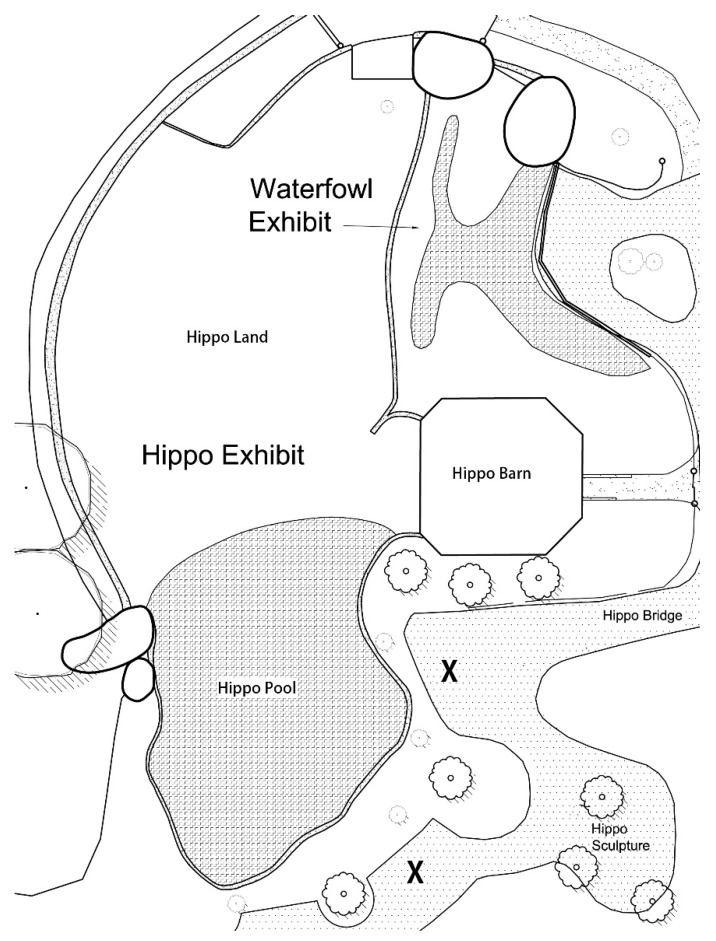
Hippo exhibit, with the two observational areas a hippo could be during the study (hippo pool and hippo land), observation points for all behavioral observations (marked ‘X’), and the indoor hippo barn (night enclosure) and surrounding areas.

**Figure 2 animals-10-01022-f002:**
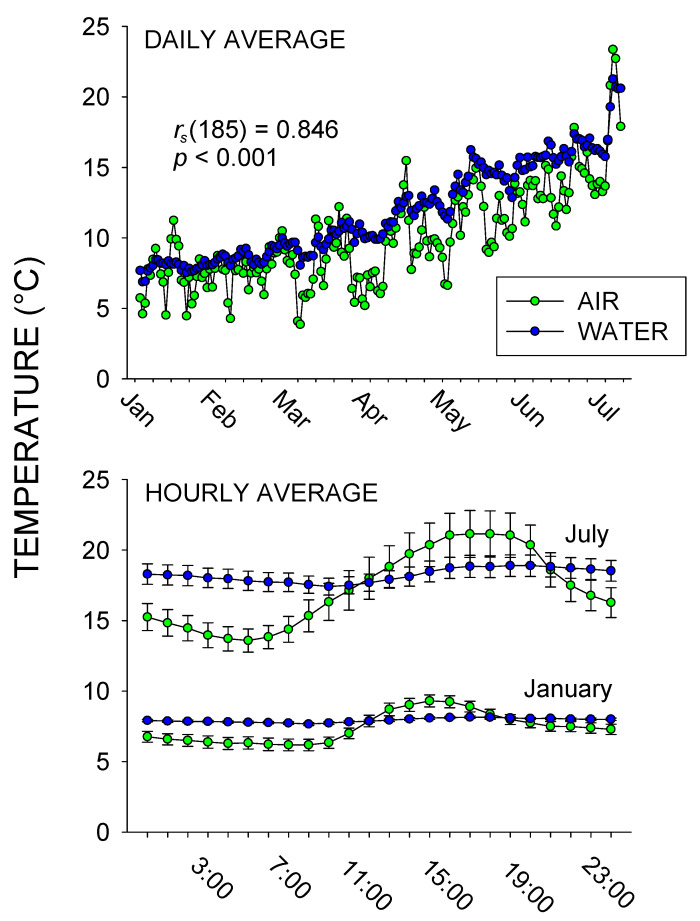
Average daily water and land temperature (top graph) and the average hourly water and land temperature (with standard error of the mean bars [SE]; bottom graph) for January vs. July. The water temperatures are the combined surface + deep water temperatures recorded. Spearman’s *r_s_* is reported for the correlation between daily air and water temperatures during the entire study (January–July; *n* = 187 days).

**Figure 3 animals-10-01022-f003:**
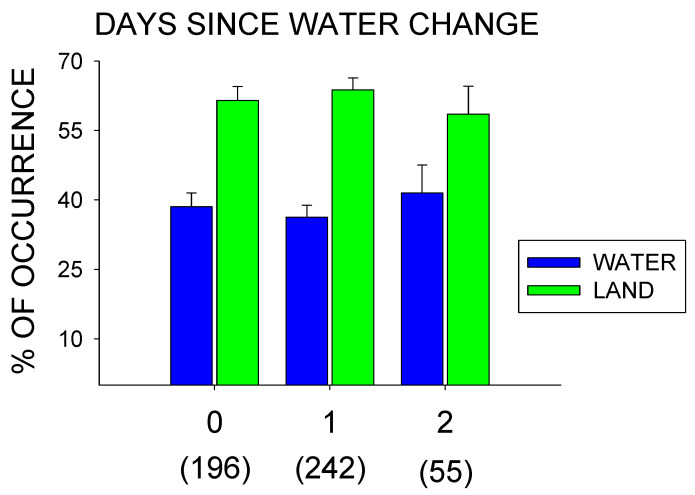
Percentage of water vs. land use (with standard error of the mean bars [SE]) based on the days since a water change (0, 1, or 2 days) for the total observations (*n* = 493) during the entire study (January–July). Number of observations for each condition are listed in the parentheses below each condition.

**Figure 4 animals-10-01022-f004:**
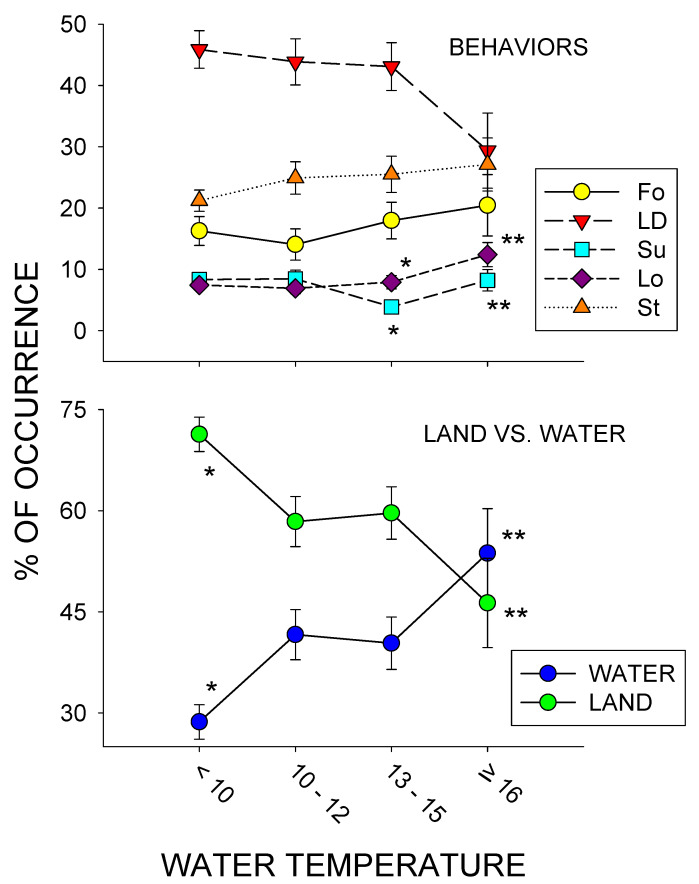
Percentage of activity (with standard error of the mean bars [SE]) for the 5 most frequently occurring responses (Lying Down (LD), Standing (St), Foraging (Fo), Locomotion (Lo) and Submerged (Su); top graph) and for water vs. land use (bottom graph) across four different water temperature conditions (<10, 10–12, 13–15, and ≥ 16 °C) for the entire study (January–July). Both graphs include standard error of the mean bars, with *** and **** indicating significant differences within each category (e.g., *** to **** = water use between <10 and ≥16 °C) across the different temperature conditions (*p* < 0.05).

**Table 1 animals-10-01022-t001:** Behaviors and definitions for each response categorized in the ethogram.

Behavior	Definition
Manipulate (Ma)	Manipulating any non-food item, such as potential enrichment devices.
Foraging (Fo)	Direct mouth contact with any food item, usually provided by the caretakers.
Lying Down (LD)	Lying down or sitting.
Submerged (Su)	Hippo is completely under the water.
Locomotion (Lo)	Walking, either forward or backward, non-repetitively.
Interacting with Other Hippo (IOH)	Any social interaction with another hippo (e.g., oriented to another hippo and vocalizing, or contacting another hippo).
Standing (St)	Standing with no locomotion.
Other (Ot)	Hippo engages in a behavior that does not meet the above behaviors (e.g., “out of sight”).

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
