# Peer review of "Activity and Pool Use in Relation to Temperature and Water Changes in Zoo Hippopotamuses (Hippopotamus amphibious)"

_animals, 2020, doi:10.3390/ani10061022_

Round 1

Reviewer 1 Report

General comments:

This is an interesting study. It is important that knowledge gaps are filled on species that are infrequently studied in zoos. At the moment, this paper is not of a sufficient standard for publication. There are numerous issues throughout and I have provided specific comments below on what should be improved.

Specific comments:

L58. Include the aims statement at the beginning of this paragraph rather than the end. Also include hypotheses.

Figure 1. Add to the figure where the observers were observing from.

L101. It’s not so usual to include a separate materials section. Generally this information is most useful merged into the other sections where the equipment is being used.

L101. Check the journal guidelines for listing equipment. Generally full details are required on the model, manufacturer and location of the manufacturing company.

L103. Provide some details on the temperature loggers – what is their range, precision etc?

L112. What does ‘casing-housed’ mean? Are they in Stevenson screens? More details should be provided as there could be implications for the accuracy of the temperature measurements.

L114. Interesting choice to put it in a boomer ball – was it not disturbed by the hippos?

L120. It doesn’t bother me but some researchers are very particular that ethogram should only refer to a complete list of the species’ behaviour, while in this context it might be better described as a catalogue.

Table 1. Not sure it is correct to have “for each response” in here as many of the behaviours are general and not in response to a particular stimuli.

Table 1. The description for foraging doesn’t need “usually provided by the caretakers” if the eating of any food is classed as foraging.

Table 1. Why was sitting in included in lying down? It seems odd because the behaviour is then mis-labeled, and this may have implications for interpretation of the results.

Table 1. The submerged category seems odd to me because until the hippos are completely submerged (as per the definition) their behaviour would be classed as ‘other’. This would miss key behaviours and impedes the aim of the study.

Table 1. All social behaviour was combined in the catalogue. Was any agonistic behaviour observed?

L129. So 30s scans were used for 30 min sessions? I couldn’t see the details for how many sessions were conducted each day? It would be good to know how the observations are balanced across the day.

L136. Is the number of total observations correct? It would seem that doing 30s scans on 3 hippos for 246 hours would be much higher than that?

L130. The section starting with “These observations were then” belongs in the statistical analysis section.

L138. More detail needs to be given on the observers. Were they experienced with hippo behaviour? Were they staff/volunteers? What training did they receive? Were they blind to the aims of the study? What reliability testing was carried out to ensure the observers all scored in a similar manner? Please add these details in.

L143. Describe how the assumptions were checked for the Pearson’s correlation run between air and water temperatures

L144. The statistical analysis of the locations and behaviours is not clear. If the statistics have been conducted including all three of the hippos, then the repeated measures nature of the sampling has not been accounted for (and running statistics with three individuals would not be robust). If each hippo has been analysed separately then that has not been made clear in the description.

L144. The behaviour doesn’t seem to have been compared against air temperature. Was there a particular reason this wasn’t explored?

L172 and elsewhere. It isn’t enough to state that a difference was significant or non-significant. Full details need to be included (e.g. the test statistic, df if applicable, and p values) as has been done at L184.

Figure 4. The figure title needs to include the behaviour abbreviations next to the behaviours listed so that it is easy to make sense of the figure.

Figure 4. The * showing the signficant differences aren’t clear here. It isn’t evident where the difference lies. Usually the star would be over a bar between the two categories that are different, or a letter system would be used.

Figure 4. It isn’t clear what the Top title (Temp (C) Bins) means?

L201. In some places the writing seems a little casual, which may be the authors’ preference. For example, it is odd to start the discussion section with “Not suprisingly” because it implies on-going discussion of something that is already being discussed. It wouldn’t be my style of writing, but given that it is not confusing or misleading then it doesn’t matter that much.

L212. The proportion of movement stated here isn’t so similar to what was found in the study. Also, it should be recognised that the activity budgets in the wild may be 24hr, not just daylight observations in this study.

L218. The information about the spike in pool use at 3pm isn’t included in the results, and should be if it is interesting enough to be discussed in the discussion.

L227 “behaviours observed by the hippos”? Maybe “shown” rather than “observed”. This sentence should be re-written anyway as it isn’t clear in it’s meaning (I assume that you mean the time since a water change didn’t affect these behaviours/activities but this isn’t clear currently in how it is written).

Section 4.3 should include discussion on previous literature. If it can’t be found for hippos then there should be some other species where this has been looked at.

L242. Locomotion doesn’t need a capital

L242-243. So the >16degC condition was statistically different from all of the other temperature conditions?

L244-246. It is quite odd though to compare land-based behaviours with water temperature. It would be more sensible to compare with air temperature.

L263. “and thus” would be more correctly changed to “as a proxy for”

L267. The conclusion should simply summarise the study. It shouldn’t include new information or discussion as has been done here with the information on training interactions

L198-259: The discussion is very light on literature and this should be improved.

L271. This is not a welfare study, nor has welfare been discussed at all in light of the resulrs so it should either be included to a sufficient degree, or this statement should be removed

L15 and L24. Correlations weren’t used for comparisons between behaviour/location and water temperature so this should be corrected

L17. Only two explanatory variables were investigated so it is too much of a stretch to say that it is the ‘most’ important factor. Also, water quality may indeed be a very influential factor but the water changes here were pretty frequent.

L21. Actually I didn’t see any comparison in the results between air temperature and how much time they spent in the pool

Abstract: Does Animals require some statistics to be included?

L28. Welfare is not discussed. This should be changed to simply “behavior of hippos”. Similarly, animal welfare is included as a keyword and should either be included in the paper or should be removed as a keyword

L36. Ref needed after Africa

L40. From “This nocturnal” to the end of the paragraph – it’s not clear why this is relevant for the current study

L50. If including a statement that a study was conducted, it is important to include information about that study (e.g. what did Blowers et al find out about social structure and exhibit use? And what did Fazal et al. Find out about introducing a female and male hippo?).

Additional: The introduction is very brief to the point of being insufficient, and should be substantially improved in breadth and depth.

Additional: There is no ethics statement. If ethics was obtained please include details of the approval. If ethics was not required, please include a statement outlining this.

Author Response

Summary/Abstract:

Adjusted to show that water temperatures were correlated with the behaviors observed and days since the pool water was changed. 

L17: Changed to 'the more important factor'.

L21: Changed to only read 'the pool water'.

The final paragraph of the Abstract now reads, "The results are discussed in terms of how these findings relate to wild hippo activity, current knowledge of zoo-housed hippo welfare, and future directions for zoo-housed hippo welfare and research."

Ethics: 

The Animals template does not include an ethics section. Prior approval from the study via the zoo and through the University of Washington (IACUC approval) are noted in the Cover Letter.

Introduction:

L36: Reference moved to sentence after "Africa".

L40-50: The Introduction has now been expanded to be more clear in both what is described and several studies done on zoo-housed hippos. For instance, the paragraph now reads: "In zoos, the regular activity of exhibited hippos is less clear. While past publications have focused on the exhibit design and care of zoo hippos and other ungulates (Forthman, 1998; Forthman, McManamon, Levi, & Bruner, 1995; Maple & Perdue, 2013), only a handful of studies have examined the behaviors of zoo-housed hippos. Blowers, Waterman, Kuhar, and Bettinger (2010; 2012) examined both the social structure and exhibit use of zoo hippos, demonstrating preferences for both familiar individuals and water areas within an exhibit. Fazal, Manzoor, Shehzadi, Pervez, and Khan (2014) examined the behavioral effects of introducing a female hippo to a solitary male hippo, showing increased activity as a result of the introduction. Little else has been done to quantify what hippos do in their zoo exhibits. A recent study suggests that North American zoos may not be adequately meeting the behavioral welfare of their hippos (Tennant et al., 2018). For instance, Tennant et al. found that contrary to wild hippo pod sizes of ≥ 10 and their regular nocturnal foraging activity, only a third of the zoos surveyed housed their hippos in groups of 3 or more and almost half of the facilities surveyed limited nocturnal foraging opportunities for their hippos. Thus, the need for more empirical information on hippo activity in zoos is clear."

The final paragraph of the Introduction now reads as follows, "The following study examined the effects of water changes and temperatures on the activity of three zoo-housed hippos. Prior to the implementation of the study, it was hypothesized by care staff that both water changes of an outdoor pool and seasonal temperature variation affected the pool use and general activity of the exhibited hippos. We, therefore, examined the effects of (1) days since the water was changed in the outdoor pool, and (2) air and water temperatures at the exhibit on (a) several possible behaviors observed, and (b) pool vs. land use within the exhibit. The purpose of the study was to document these behavioral effects, which would allow us to empirically evaluate the behavioral welfare of the exhibited hippos, as well as suggest future directions for the assessment and care of these and other zoo-housed hippos."

The Introduction is otherwise complete. There are very few published studies (3) on the behavior of zoo-housed hippos. The welfare implications of the following study are discussed, as well as potential future studies, in the discussion and conclusions. If there is any other information the reviewer would like included in the Introduction, we are open to suggestions.

Methods:

L101: Materials section is now listed as 2.1.1. We wanted to distinguish this to account for the particular materials included since the temperature data loggers were important.

Other Materials information now includes a website for the data loggers, model #, IP waterproof rating, and range.

L112: We removed 'casing-housed' from the description; these were the casings designed by Thermoworks for the data loggers (listed in the Materials section, now with model #), and were designed to not interfere with any temperature readings. Since they are listed in the Materials section, it is redundant to state that data loggers were used with their casings.

L114: The easiest way to record surface water temperature without interference by the hippos was to place the data logger inside a boomer ball too large for the hippos to otherwise disturb. The boomer ball was regularly used for the hippos, but during this study, it was not used as an enrichment device and maintained throughout the study with the data logger sealed inside. The boomer ball would still fill with water and float, so the data logger always floated inside the boomer ball at approximately the same shallow level of water. The hippos were never observed to do more than making incidental contact with the boomer ball. 

L120: Because of the inclusion of the "Other" category, this is an exhaustive list of behaviors that could be observed. It may not be a long list of possible behaviors, but it operationally defines each response into a quantifiable, mutually exclusive, and exhaustive set of responses, and thus meets most definitions of an ethogram. 

Table 1. Using 'for each response' is correct, in the general sense. They are definitions for each response, and responses need not be defined in reaction to particular stimuli. For the 'Foraging' response, 'eating' was changed to 'direct mouth contact with', and we specified that Foraging would usually occur to food provided by keepers, but could also include consuming anything edible that was in the exhibit. Lying Down included sitting because the animal could be only partially on the ground. Less common in hippos, more common in other animals that the same research assistants might observe. But still possible for a hippo, thus important to note. "Submerged" was an important behavior to observe because you could know a hippo was in the pool (i.e., the observer just recorded them in the pool), but not know what they were doing in the pool. So you still recorded the location, but not necessarily the activity. Finally, almost all interactions observed were incidental contacts. We never observed any aggressive or otherwise agnostic interactions, and only observed IOH for 0.22% of all observations. 

L129: We included a statement that there were 1-8 observations per day across the 179 total days of observation (493 total observation sessions). And yes, they were 30s instantaneous time (pinpoint) samples for half-hour sessions. 

L130: the 'These observations' statement explains how the data were aggregated before any statistical analyses were applied. It makes the most sense to leave it here, where how data was collected and organized is described.

L136: 'observations' was changed to 'sessions', since it's 493 total sessions (over 5,000 individual observations) made across the 6 months of the study.

L138: Descriptions of the research assistants, along with the training, are now provided. Also, yes, they did not know when a dump occurred, but they had a general sense of the temperature at the exhibit.

L143: We are not clear what assumptions you would like described in terms of being checked for a Pearson's Product Moment Correlation. They are temperature recordings, so interval measures, they were independently sampled, and the graph shows their linearity. We believe this is already presented in a straight-forward manner without describing any assumption requirements.

L144: We chose to compare behaviors to water temperatures because they were the more stable of the temperature measures. This is why we presented the air and water graph, and this is noted at the beginning of the Discussion section.

L144 (continued): As noted previously, all data were averaged per session for all hippos across all behaviors. Meaning, there were no independent hippo analyses, so as to not inflate the significance of any findings by multiplying # of hippos with # of sessions per condition. Our 'n' is simply the number of sessions per condition. 

This is addressed more directly by Swaisgood and Shepherdson (2005) when discussing how to apply statistics to relatively small numbers of subjects via repeated measures (e.g., number of days) for any one condition. We choose to use one of the most conservative methods to address this approach, which is to average all responses across all subjects within each session.

Results:

Line 172: We've included the non-significant test results for the days since a water change (0, 1, and 2) compared to pool use. However, APA-style generally does not recommend listing all non-significant results unless directly relevant. We have compromised with the reviewer's suggestion and included the above for its relevancy.

Figure 4: Abbreviations for each behavior are now listed in the figure caption. * and * have been changed to a, b, c, and d (italicized) to make significant differences more clear. The 'Temp Bin' description was removed from the top, as this was unnecessary. 'BEHAVIORS' and 'LAND VS WATER' have been added as smaller titles into each graph. 

Discussion:

L201 (start of Discussion): Changed "Not surprisingly" to "As expected". While still a bit cordial, we do want to make it clear to the reader that finding a strong correlation between the air and water temperature at the hippo exhibit was par for the course. 

L212: The overall activity pattern is similar. Of course, locomotion itself was less similar, given the captive restriction (i.e., our hippos did not have to travel to get food), but that can be inferred from the results and is less relevant to the purpose of this manuscript.

L218: We changed this to 'an unreported spike'. It was interesting to see this trend but was also unrelated to changes in water temperature or pool dump. As such, it's still worth mentioning for the purposes of future studies that might find a link between appetitive behavior and water use in hippos. But, we wanted to limit the discussion to a brief mention for that purpose, and not as a major distraction from the relevance of this study. 

L227: We changed the entire paragraph to read, "No significant differences were observed in either pool/land use or general activity as a result of the amount of time (days) since the pool water was changed. This result differed from anecdotal reports by the care staff, who suggested that the hippos used their pools less as more time passed since a water change. This was also one of the primary reasons for examining the effects of water changes on both the hippos’ general activity and their overall pool use."

Section 4.3: The simple answer is that there are 3 published behavioral studies (plus one survey) conducted on zoo hippos. We could find no other studies on pool use in relation to water change and/or water quality in any captive species. If there is literature the reviewer would like to suggest, we would welcome such suggestions.

L242: Locomotion no longer capitalized, as is now the same for all other categorical labels mentioned in the discussion.

L242-243: The >16 condition was significantly different than the 13-15 condition for two of the behaviors and the <10 condition for pool and land use, as noted in the graph and the results. 

L263: Changed to "and therefore".

We are otherwise open to specific edits, including reference suggestions, that reviewer #1 would like to offer. 

Reviewer 2 Report

The manuscript of Fernandez et al. investigated the relationship between water changes and temperature with behavioural activity and pool use in zoo-housed hippopotamuses. The manuscript is well-written and the results present new details on behaviour of zoo hippos which can provide better insights to increase the welfare of this species in captivity. However, I would like to suggest some revisions before publication:

  1. P1, L 28: The conclusion of the abstract is too concise, I would suggest to include a couple of lines where the authors briefly mention these implications.
  2. P2, L 46: Please, consider to change ‘past writings’ with ‘past research’.
  3. P2, L 49: Please, consider to modify ‘in humane care’ with ‘captivity’.
  4. P2, L 49-51: I would suggest to include the references in the text as ‘Blowers et al. (2010; 2012)’ instead of writing all authors’ names. Same for the reference ‘Fazal et al. (2014)’. Moreover, the journal guidelines say that references must be placed in square brackets and as a number, according to the order of appearance. (https://www.mdpi.com/journal/animals/instructions#preparation).
  5. P2, L 56: Can the authors better elaborate (supported by appropriate literature) why they decided to focus on air/water temperature, water changes and pool use in their study? Would this contribute to improve the welfare of zoo-housed hippos? (e.g. the water is a form of thermoregulation in nature). This will help the readers to understand the importance of doing research on this topic.
  6. P2, L 59-62: Please, consider to move this info in materials and methods section.
  7. P2, L 62-63: It seems to me that these sentences are repeating the same concept of L 58-59.
  8. P5, L 129-130 and 136: How many sessions per day and observations per h/per day?
  9. P5, L 138: I am a bit concerned about the high number of observers (n=33) employed for the data collection. May this have biased the data? I am sure that they were all trained, however 33 observes may be considered too many.
  10. P5, L 152: A strong ‘positive’ correlation?
  11. P5, L 153: Is ‘185’ the number of days of data collection? If so, why is this 185 and not 187 as in L 138?
  12. P6-7, L 166-189: The authors state that they did a non parametric analysis (Kruskal-Wallis test on ranks), thus I would expect to see the results presented by median and IQR instead of mean and SE.
  13. P8, L 1205: Please, consider to modify ‘activity’ with ‘activities’.
  14. P9, L 217: I would suggest to add an appropriate reference instead of writing ‘see below’.
  15. P9, L 217-219: This is interesting! The authors discussed a bit about this topic whereby I would suggest to include this data in the results section. Did the authors analyse this data? Is there any significant result?
  16. P9, L 232-237: Is there any other study that can be used to confirm this assumption? Perhaps, studies on other zoo-housed species where there was a link between water quality and time spent in the pool?
  17. P10, L 267: Please, consider to modify ‘researchers have used’ with ‘studies investigated the use of’.
  18. P10, L 269: Did the authors mean this: ‘particularly how factors such as environmental enrichment may be related to the amount of time hippos spend in enclosure water areas and/or pools’?

Author Response

Edits are as follows, according to the suggested revisions:

1. The final line of the abstract was changed as such, "The results are discussed in terms of how these findings relate to wild hippo activity, current knowledge of zoo-housed hippo welfare, and future directions for zoo-housed hippo welfare and research."

2. Changed to "past publications", since some of the citations referenced (e.g., Forthman, 1998) are not based on research studies. 

3. Changed to "zoo-housed hippos." One of the authors tries to avoid using the word 'captivity' when describing animals in zoos.

4. We used APA-style referencing for the manuscript since we find it easier for ourselves, the reviewers, and the editors to use the reference list. APA-style suggests using all authors' names (up to 5 authors) for the first time a reference is mentioned. After the review process, we will adjust the reference list and citations within the text according to the standard formatting. 

5. Based on Reviewer #3's comments, we adjusted the final sentence of the paragraph as such, "Little else has been done to quantify what hippos do in their zoo exhibits. A recent study suggests that North American zoos may not be adequately meeting the behavioral welfare of their hippos (Tennant et al., 2018). For instance, Tennant et al. found that contrary to wild hippo pod sizes of ≥ 10 and their regular nocturnal foraging activity, only a third of the zoos surveyed housed their hippos in groups of 3 or more and almost half of the facilities surveyed limited nocturnal foraging opportunities for their hippos. Thus, the need for more empirical information on hippo activity in zoos is clear." Additional information about the purpose of the study (the use of pools and their behaviors) is addressed in points 6-7.

6-7. We've adjusted the final paragraph of the Introduction so that it reads more clearly as to the purpose of the study. We maintained this paragraph, albeit with less methodological information, because it's standard research manuscript format to have the last paragraph of an Introduction detail the purpose of the current manuscript. The paragraph now reads as follows, "The following study examined the effects of water changes and temperatures on the activity of three zoo-housed hippos. Prior to the implementation of the study, it was hypothesized by care staff that both water changes of an outdoor pool and seasonal temperature variation affected the pool use and general activity of the exhibited hippos. We, therefore, examined the effects of (1) days since the water was changed in the outdoor pool, and (2) air and water temperatures at the exhibit on (a) several possible behaviors observed, and (b) pool vs. land use within the exhibit. The purpose of the study was to document these behavioral effects, which would allow us to empirically evaluate the behavioral welfare of the exhibited hippos, as well as suggest future directions for the assessment and care of these and other zoo-housed hippos."

8. Adjusted the sentence to include, "(493 total observations across the 179 days [1-8 observations per day] for 246.5 hours of behavioral observation)."

9. The statement now includes the following sentence about the training of these research assistants: "Observers were typically registered for independent research credit through the Psychology Department at the University of Washington (PSY 499) and received observation training by live training sessions at the beginning of each semester and weekly lab meetings throughout the study."

10. 'positive' now included.

11. 187 is the total n. 185 is the degrees of freedom (df; n - 2). 

12. While non-parametric tests create ranks and then compare median scores of those ranks, it wouldn't make much sense when discussing how much we observed any activity in terms of median scores or the ranks given to those scores. Just because we rank a score due to a lack of normality and/or homogeneity of variance so as to be able to run an inferential statistic, doesn't mean we want to present those findings in terms of those ranks. Mean and SE scores of those means are still very much relevant. 

13. Changed to, "the behaviors observed". 

14. Adjusted with the reference, followed by 'see below'.

15. We included 'an unreported spike', and while it is a neat effect, and therefore we wanted to briefly mention it, the results had no other implications in terms of seasonality or time since a water change. It was just that the hippos, before that large 3 pm feed, would get in the pool more the hour before the feed, and then come out on land while eating. We have no idea why they did this (we suggest at least two possibilities), and it certainly would be interesting to further examine, particularly if you could push around swimming time and/or amount of swimming by changing how and when you feed the hippos. It's better left as an interesting observation worth some future research investigating. 

16. Not that we know of. There are 3 published studies on hippo behavior of any kind done prior to this study. There are no studies that we're aware of that have examined the relationship between water quality and water use by any captive species/animal. However, if you're asking how safe is the assumption that the water quality was directly affected by the amount of time since the water had been changed, yes, it's a very safe assumption. Hippos are large and they defecate a lot and everywhere, spreading their feces across any area, land or water, by wagging their tails rapidly while defecating. We highly recommend you search Youtube for videos of captive hippos defecating to get an idea of how safe this assumption is. 

17. Changed accordingly. 

18. Changed sentence to read, "particularly how factors such as environmental enrichment relate to the amount of time hippos spend in enclosure water areas and/or pools."

Reviewer 3 Report

This study examined the effects of water quality and temperature on the behaviors and land-use in captive hippos. The study is overall sounds in terms of study designs and protocols and the topic matches the theme of the special issue. I have several comments for further clarification.

1) P1, L40-42: What kind of the human-hippo conflicts occurred because of the nocturnal foraging?

2)P2, L52-55: I agree that it is important to consider the nocturnal foraging opportunities for captive hippos. However, this study did not collect data at night. Therefore, I do not understand why the authors put emphasis on nocturnal activities in the first two paragraphs of the introduction. I also agree “the need for more empirical information on hippo activity in zoos is clear.”, but again I do not understand the link between the previous sentence and this remark. It may be better to include more explanations.

3) Most of the data in this manuscript discusses the temperature and water quality. It is better to include such relevant information in the introduction. For example, is there ideal range of temperatures for captive hippos discussed in any manual? The range of temperatures in the habitats of wild hippos? Or lack of such information?

4)P4, Table 1: It looks like the definition of “Foraging” is incomplete. For example, manipulating food items was included in this category?

5) P5, L 138-139: A lots of observers were involved in this study. Needs more information regarding the consistency of the data collection (e.g. inter-observer reliability or training on these observers)

6) P 9, L217-219: It is interesting but I do not find the data in the results for this discussion.

Author Response

Responses to comments below:

1. Included in the text, after noting human-hippo conflicts, "(e.g., crop destruction and physical threats)".

2. We agree this did seem a little confusing. The point was meant to summarize the findings of Tennant et al. (2018), as well as illustrate how current practices do not match how hippos exist in the wild. As such, we've revised the statement to read: 

"A recent study suggests that North American zoos may not be adequately meeting the behavioral welfare of their hippos (Tennant et al., 2018). For instance, Tennant et al. found that contrary to wild hippo pod sizes of ≥ 10 and their regular nocturnal foraging activity, only a third of the zoos surveyed housed their hippos in groups of 3 or more and almost half of the facilities surveyed limited nocturnal foraging opportunities for their hippos. Thus, the need for more empirical information on hippo activity in zoos is clear."

3. Unfortunately, no, there is no given ideal range for captive hippos, certainly not one that is empirically based. And, in the wild, their range covers from as far south as South Africa to as far north as Sudan. In fact, Noirard et al. (2008), as mentioned in the Discussion, examined some of this dramatic variation in the daily and seasonal temperatures in Niger, and how this effected hippo water and land use as a test of thermoregulation. The simple answer is, while it's an interesting question, we treated this topic inductively in terms of what the air and water temperatures were and how that correlated with their pool and land use. We discuss it further in the Discussion, but I'm not sure we can say much in the Introduction that really adds to the conversation. 

4. We changed "Eating" to "Direct mouth contact". So, yes, any contact with an edible item was considered foraging.

5. We added the following sentence, "Observers were typically registered for independent research credit at the University of Washington (PSY 499) and received observation training by live training sessions at the beginning of each semester and weekly lab meetings throughout the study." While I (the first author) typically like collecting some form of IOA, I found that when working with a large number of research assistants, as I did throughout my postdoctoral research, the easiest way to maintain valid measurement is to spend half the lab time each week going over the ethograms for all our projects. There's bound to be both some not-so-great observers and some drift, but when you monitor both the observers and the data closely, you catch that pretty immediately and rectify it then, rather than assess the problem via IOA at the end of the study, when it's too late to do anything about it. 

6. We did not show results for the hour-to-hour pool vs. land use since it really wasn't relevant to either the dump or temperature results. However, we did want to note that we did see that spike in pool use activity just prior to the 3 pm land feed. It's one of those, "hey, someone down the road might have a study or result related to this" kind of comments. For now, we've noted that it was an unreported spike. We are more than happy to remove the comment if the reviewer finds that preferable. 

Round 2

Reviewer 1 Report

Title: Activity and Pool Use in Relation to Temperature and Water Changes in Zoo Hippopotamuses

Thank you for your revised manuscript. There are still a few remaining points to address. Most of these were outlined in the previous review but were only partially addressed:

Original comment

New comment on this issue

Provide some details on the temperature loggers – what is their range, precision etc?

This was only partially addressed. Please add specifications on the loggers. The company lists relevant details on range (both min and max) as well as accuracy on their website for this product.

What does ‘casing-housed’ mean? Are they in Stevenson screens? More details should be provided as there could be implications for the accuracy of the temperature measurements.

This was only partially addressed. The reason that I asked about whether the casing acts like a Stevenson screen is because direct sunlight affects the accuracy of temperature loggers. Usually they are placed in a Stevenson screen to prevent this. It is still not clear whether that is the case for this logger and its casing. 

The submerged category seems odd to me because until the hippos are completely submerged (as per the definition) their behaviour would be classed as ‘other’. This would miss key behaviours and impedes the aim of the study.

My point for this original question was that the ethogram contains no in-pool activity except submerged. So if the hippo was swimming but not submerged, their location will be classed as in the pool, but their behaviour will be classed as other. Given the aims of the study, could the authors make some comment about why non-submerged pool activity wasn’t included as a behaviour in the ethogram

More detail needs to be given on the observers. Were they experienced with hippo behaviour? Were they staff/volunteers? What training did they receive? Were they blind to the aims of the study? What reliability testing was carried out to ensure the observers all scored in a similar manner? Please add these details in.

This has been partially addressed, but crucially what is still missing is an assessment of data quality collected by these observers. A very large amount of observers were used. No reliability measures for the observers have been presented (presumedly because they weren’t checked), which is standard practice for measuring behaviour through observation. This potentially undermines the validity of the study so the authors should address this issue by outlining how they can be confident in the data collected by these students.

Describe how the assumptions were checked for the Pearson’s correlation run between air and water temperatures.

The author in their response have described some assumptions in their response only (not in the manuscript) but have not mentioned normality (this has only been mentioned for the behavioural analyses), influential outliers, and homoscedasticity. It should be stated explicitly which assumptions of the models were checked and how.

It isn’t enough to state that a difference was significant or non-significant. Full details need to be included (e.g. the test statistic, df if applicable, and p values) as has been done at L184.

The authors have added the full statistics at this point but not at other points, citing the APA reporting guidelines. The APA guidelines https://apastyle.apa.org/jars/quantitative that I checked here state that full statistics should be provided when testing hypotheses (as in this study). It does not make any distinction between significant and non-significant results. These details are important for transparent results, especially given differing opinions on the use of p-values at all. Please include them in each spot within the results that you claim there is a significant or non-significant difference (unless that information is provided somewhere else such as a figure). For example, this information is needed at the end of L187.

Figure 4. The * showing the signficant differences aren’t clear here. It isn’t evident where the difference lies. Usually the star would be over a bar between the two categories that are different, or a letter system would be used.

The * were changed to letters but the meaning is still not clear because they haven’t been used in a conventional way. When using letters to show statistical difference, different letters are used to show that two categories are statistically different from each other within the same variable. The current figure is still not clear because it suggests (for example) that for ‘Water’ behaviour the <10 and the >16 categories are statistically the same because they have the same letter. See here for an example of where subscript letters have been used correctly: https://journals.plos.org/plosone/article/figure?id=10.1371/journal.pone.0008263.g002

The conclusion should simply summarise the study. It shouldn’t include new information or discussion as has been done here with the information on training interactions

There is still new information included here (e.g. there is a new reference introduced here on penguins) which should be moved to the discussion and this section used to summarise and conclude the current study.

L15 and L24. Correlations weren’t used for comparisons between behaviour/location and water temperature so this should be corrected

The word correlation is still used although the data was not analysed using correlations, which is misleading for the reader. Change to a suitable word such as ‘associated’

L28. Welfare is not discussed. This should be changed to simply “behavior of hippos”. Similarly, animal welfare is included as a keyword and should either be included in the paper or should be removed as a keyword.

This has been partially addressed because some welfare information has been included in the introduction. But this is quite superficial and it is still not addressed at all in the discussion. Clear links need to be drawn between behaviour, welfare, and environmental needs for hippos such as pools in zoos. For example, consider why behaviour is a useful measure for welfare (which in itself is a vastly multi-dimensional science not restricted to behavioural measures). Perhaps check out https://doi.org/10.1078/0944-2006-00122 on this issue.

Why specifically is naturalistic behaviour going to be useful for welfare? (Perhaps look at recent papers discussing this: https://doi.org/10.1080/10888705.2019.1672552 and https://doi.org/10.3390/ani9060318 )

An alternative approach could be think of the behaviour from a pure ethological interpretation (re: Tinbergens 4 whys – why is pool related behaviour performed by an individual (causation & ontogeny) or species (adaptive value & phylogeny). These questions are relevant to welfare because they relate to motivations, reproductive fitness, health, and survival etc etc.

My overall point here is that while there are few zoo studies specifically on hippos, adequate discussion of literature is not just about the specific species, but making clear links between broader concepts particularly if making claims that the study is important to welfare. Currently it is not clear whether the authors are very familiar with welfare as a science.

The alternative is to drop the conclusions about welfare and present it simply as a behavioural study.

There is no ethics statement. If ethics was obtained please include details of the approval. If ethics was not required, please include a statement outlining this.

Authors state that ethics was included in the cover letter, however ethics approval details need to be included in the paper itself. It should be placed in a suitable place in the methods.

Author Response

Details of the data loggers in terms of their range and accuracy are now presented as such: "The data loggers had an IP67 waterproof rating with a range from -35 to 80°C (±1°C)." 

The cases have now been described as "three protective metal cases" rather than casings. Thermoworks describes the cases as "waterproof", although that's a misnomer (the data loggers themselves have an IP67 rating, as noted above).

We cannot provide any other detail about the cases, other than Thermoworks assured us that they would be suitable for our purposes and would not interfere with the IP rating or accuracy of any temperature readings. The design and maintenance crew that installed the devices at the start of the study specifically placed the data loggers in the cases at points in the exhibit that would not receive direct sunlight, which was of most concern for the device fixed to the tree since the other two devices were not in direct contact with sunlight or the elements. We've reworded the methods to reflect this with the following: "the three temperature data loggers were placed at three different points of the exhibit that did not receive direct sunlight: fixed to a tree above the exhibit and out of reach from both visitors and the hippos, and that was shaded by leaves and branches..."

All of the 8 responses could be recorded as such, and only as one response (mutually exclusive) in the pool. For instance, a hippo could be engaged in locomotion or standing in the pool just as it would be on land. The "Submerged" response was only possible in the pool and was recorded as such because it allowed observers to still record where the hippo was without knowing exactly what the hippo was doing. This only occurred for ~7% of all behaviors recorded, since most of the time, an observer could record what a hippo was doing while submerged (i.e., they could see it locomoting or standing). We have added the following sentence to make this more clear: "All 8 responses could be recorded in the pool, and 7 of the 8 responses could be recorded on land, with the exception of the “Submerged” response, which identified a hippo in the pool without the ability to identify the exact response. " 

To account for the accuracy of the data, the following sentences have been added to the Methods: "Observations were examined weekly by the first author for consistency across all observers, and drift was accounted for during these weekly checks, as well as through weekly lab meetings. Observers were blind to the conditions of the study, although they had a general sense of the weather conditions during their observation sessions."

Just as an additional note, it's fairly uncommon for continuous  (focal) measurement or instantaneous time (pinpoint/scan) samples to be tested for Interobserver Agreement (IOA). Almost all forms of IOA are performed instead on interval (one-zero) samples or discrete trial recordings because with either of those methods, you can generate consistent recording intervals that match each other. I (the first author) have generated IOA through total agreement, percentage agreement, and Kappa on pinpoint samples, and it's incredibly difficult to do since it requires making sure two observers start their observations at the exact same time and from the exact same observation viewing points. To ensure reliable, valid behavioral recordings, particularly when working with large sets of data (6+ months) and many observers (20+), I've found the best way to do this is to train research assistants, then train them again each and every week, and spend every week scrutinizing their data to make sure the data is consistent across the other observers for that week. 

In terms of Pearson's correlation, it turns out, we missed testing the temperatures for normality (Shapiro-Wilk), in which the data failed. Therefore, we recalculated the data according to a Spearman's rank correlation coefficient, which does not require the normality assumption. In addition, Spearman's correlation requires a monotonic data set, which is shown by the graph. The result was a slightly lower correlation coefficient (.846 instead of .883), but that otherwise is still significant at p < .001. The wording in the Methods and the graph have been adjusted accordingly. 

All non-significant results are now reported in the manuscript.

In terms of our graph, we are following the standard convention of showing what is significant via a to a, then b to b, etc. It's the same as the graph the reviewer shows, except that we do not have more than one significant difference within each category (hence, why there is no 'ab' or the like). Just as the reviewer correctly noted, within the category of 'water use', water use in the <10 and >16 conditions are significantly different from each other (thus, each being 'a'). We've tried to make this more clear by pointing out in the figure caption that d to d means a significant difference. The four significant differences are thus, a to a, b to b, c to c, and d to d.

I think the confusion may be on the part of this being a line graph, rather than a histogram, and that there is only one significant difference in each category, but it's exactly like the graph presented by the reviewer, where the top left A graph shows the D0 condition as being 'ac' because it is significantly different than the D20 condition, 'a', and D30 condition, 'bc', but not the D25 condition, 'b'. 

The Conclusions section is now just a brief review of the results, as well as the suggestion of the importance of pool temperature and pool use in managing hippo exhibits.

The Discussion has been expanded to include several more references relevant to examing welfare issues. We appreciate the reviewer's suggestions about the philosophical implications as to what welfare science is. No doubt, what we call behavioral welfare, or any form of welfare science, is at times subjective, and worth examining in terms of caretaker- and visitor-desired changes in behavior verse behavioral changes beneficial to the individual and/or species. However, our study is more directly focused on the assessment of a welfare-related issue, as is the case for most applied animal behavior studies. The simple point is that how hippos use their exhibits, specifically how they use their pools, what their general activity is, and how all the above is connected with pool temperatures is very much a hippo welfare issue.

In terms of correlation, outside of the Spearman's correlation tests, we've changed the word to either 'link' or 'relationship'. However, we still refer to the study as a correlational verse causal study, because this is using the broader sense of the term.

Finally, we've included an ethics statement that lists our IACUC #, as well as an author contribution section.

Reviewer 3 Report

The manuscript was improved from the previous version. I can now recommend this manuscript for publication.

Author Response

Thank you.

Round 3

Reviewer 1 Report

Thank you for sending the revisions. I am happy with most of them and appreciate the reviewers taking the suggestions on board (although I disagree on the IOR comments, but it is apparent that the authors are confident in their data quality, so let's leave it at that).

However, I would suggest that the authors take one more look at the figure with the subscript letters compared to the example that I sent through. In the author reply you wrote:

"I think the confusion may be on the part of this being a line graph, rather than a histogram, and that there is only one significant difference in each category, but it's exactly like the graph presented by the reviewer, where the top left A graph shows the D0 condition as being 'ac' because it is significantly different than the D20 condition, 'a', and D30 condition, 'bc', but not the D25 condition, 'b'."

Your interpretation of the example figure that I sent through is actually the opposite of what the figure actually shows (see link here: https://journals.plos.org/plosone/article/figure?id=10.1371/journal.pone.0008263.g002).

In the top left panel of that figure D0 shares a letter with D20 and D30 therefore they are not significantly different. D0 is only significantly different from D25 because they have different letters (and this is evident just from the column and error bars). D20 is also significantly different from D25 (& D30). This is the conventional way to use these subscripts, which is why I think the lettering in the current figure will confuse your readers. It is not technically incorrect given what has now been included in the figure title, however, I wanted to re-clarify the above so that you have the option to change the lettering, if you should want that, prior to publication. 

Author Response

Yes, we flipped that (or rather, I, the first author). I've changed it accordingly so that the significant difference within a category is listed as being between * and **, and the caption is clear that it is within a category (e.g., * to ** water use). 

Thank you for the time and effort you have put into these reviews. It has helped make this paper substantially better as a result.